

# Human altruistic tendencies vary with both the costliness of selfless acts and socioeconomic status

Cyril C. Grueter, Jesse A. Ingram, James W. Lewisson, Olivia R. Bradford, Melody Taba, Rebecca E. Coetzee and Michelle A. Sherwood

School of Anatomy, Physiology and Human Biology, University of Western Australia, Perth, Western Australia, Australia

## ABSTRACT

Altruism toward strangers is considered a defining feature of humans. However, manifestation of this behaviour is contingent on the costliness of the selfless act. The extent of altruistic tendencies also varies cross-culturally, being more common in societies with higher levels of market integration. However, the existence of local variation in selfless behaviour within populations has received relatively little empirical attention. Using a 'lost letter' design, we dropped 300 letters (half of them stamped, half of them unstamped) in 15 residential suburbs of the greater Perth area that differ markedly in socioeconomic status. The number of returned letters was used as evidence of altruistic behaviour. Costliness was assessed by comparing return rates for stamped vs. unstamped letters. We predicted that there is a positive association between suburb socioeconomic status and number of letters returned and that altruistic acts decrease in frequency when costs increase, even minimally. Both predictions were solidly supported and demonstrate that socioeconomic deprivation and elevated performance costs independently impinge on the universality of altruistic behaviour in humans.

## INTRODUCTION

Prosocial sentiments, i.e., caring about the welfare of others, have emerged as hallmarks of humans (*Gintis, 2003*; *Henrich et al., 2004*; *Hill, Barton & Hurtado, 2009*; *Alvard, 2012*; but see *Burton-Chellew & West, 2013*). Altruism represents a special case of prosociality in which an actor helps others at a personal cost. Altruism can become fixed in stable groups of humans and other animals through kin selection, a process whereby individuals accrue indirect benefits through the successful reproduction of relatives (*Hamilton, 1964*). Explaining altruism directed at unrelated individuals requires the framework of direct reciprocity (*Trivers, 1971*) or indirect reciprocity (*Alexander, 1987*). In direct reciprocity, the temporary costs individuals incur by performing an 'altruistic' act will be recouped by subsequent generosity on the part of the recipient. In indirect reciprocity, lending a helping hand can enhance the reputation of individuals and increase the likelihood of others cooperating with them in the future.

Corresponding author
Cyril C. Grueter,
cyril.grueter@uwa.edu.au

However, when altruism surfaces in large anonymous groups of unrelated individuals canonical evolutionary approaches based on nepotistic biases, direct reciprocity and indirect reciprocity can largely be ruled out as explanations. It has been argued that large-scale cooperation can be maintained when behaviourally rather homogenous groups of prosocial individuals gain a competitive edge against groups of less prosocially cohesive individuals (cultural groups selection) (*Henrich, 2004*; *Richerson et al., 2016*; but see *Yamagishi & Mifune, 2016*).

Despite its ubiquity in human societies, the propensity for altruism varies and is expected to be superseded by selfish motives when acts of altruism are more costly, that is when they entail larger sacrifices to one's own payoff (*Fehr & Fischbacher, 2003*). This argument has been substantiated through economic games such as the dictator game, e.g., when the cost of relinquishing one monetary unit to the recipient increases, the dictator donates less (*Andreoni & Miller, 2002*). In a study using children, it was shown that in a costly sharing game (when delivering rewards to a recipient required personal sacrifice) the likelihood of prosocial behaviour was lower than in a prosocial game (in which offering a reward to a recipient had no inherent costs) (*House et al., 2013*). Further evidence for cost-dependent variation in altruism is presented by *Stewart-Williams (2007)* who used questionnaires about help exchanged with individuals of different relatedness classes and found that with increasing costs of help, nonkin received a smaller share of the help given than kin.

Prosocial inclinations are also contingent on the social and ecological environment (*Lamba & Mace, 2011*). *House et al. (2013)* demonstrated the emergence of population-specific variation in costly prosociality during middle childhood. A cross-cultural study of behaviour in ultimatum games showed that levels of prosociality increased with market integration and the reliance on cooperative partners from outside the immediate family (*Henrich et al., 2005*; *Henrich et al., 2010*). However, the existence of local variation in prosocial behaviour *within* such industrialized populations has received relatively little empirical attention (but see *Wilson, O'Brien & Sesma, 2009*; *Nettle, Colléony & Cockerill, 2011*; *Holland, Silva & Mace, 2012*; *Silva & Mace, 2014*; *Silva & Mace, 2015*).

A simple but powerful way to quantify pure altruism toward strangers in a naturalistic setting (urban context) is through the lost letter experiment. This experiment involves dropping letters on the sidewalk and counting the number of letters that are picked up by passers-by and mailed to the addressee (*Milgram, Mann & Harter, 1965*). Previous applications of this methodology have found that letter return rates were correlated with perceived neighbourhood quality (*Wilson, O'Brien & Sesma, 2009*) and objective neighbourhood wealth and socioeconomic status (*Nettle, Colléony & Cockerill, 2011*; *Holland, Silva & Mace, 2012*; *Silva & Mace, 2014*).

In the present study, we aim to apply the lost letter technique to simultaneously disentangle the effects of both socioeconomic status and the inherent costs of executing a task on the prevalence of altruistic behaviour in an urban setting. We first predicted that spontaneous prosociality would be less prevalent in areas of low socioeconomic status because poorer neighborhoods are characterized by low neighbourhood quality (*Wilson, O'Brien & Sesma, 2009*), high crime rates (*Sampson, Raudenbush & Earls, 1997*; *Nettle, Colléony & Cockerill, 2011*), low social capital and trust (*Sampson, Raudenbush & Earls, 1997*;
*Li, Pickles & Savage, 2005*; *Nettle, Colléony & Cockerill, 2011*), and low rates of civic engagement (*Li, Pickles & Savage, 2005*). Hence letters dropped in socioeconomically poorer areas should have a lower likelihood of being returned. We also predicted that increased costs of returning the letter would decrease altruism (*Fessler, 2009*). That is, among the returned letters there would be fewer unstamped letters; due to the additional financial expense required to post an unstamped letter, it can be implied that returning unstamped letters imposes a larger cost to the actor.

## METHODS

### Data collection

We (the authors) dropped a total of 300 letters (150 stamped and 150 unstamped) were dropped in 15 residential suburbs in the Perth Metropolitan area that differed in levels of socioeconomic deprivation/affluence. We distributed twenty letters, ten stamped and ten unstamped, face up on sidewalks of each suburb. We addressed envelopes to one of the author's home address; we did not drop any letters in the suburb that the letters were addressed to. We chose the addressee's name to be 'S. Roberts,' as we considered it to be a gender-neutral name. We chose a 'Western' name to remove any potential ethnic biases (*Ahmed, 2010*). There was no 'return to sender' address. We addressed all the letters in the same handwriting in the same standard white envelope. Since the letter was handwritten, it can be deduced that the letter did not contain official documents, utility bills or company letters. The content of the letter was a folded piece of A4 paper containing the name of the suburb the letter was dropped in and a note on whether it was stamped or unstamped. The content of the letter was indistinguishable from the outside.

We dropped all 300 letters in their respective suburbs on the same evening between 17:00 and 19:00. We dropped the letters on a Saturday evening to ensure no postmen would pick up the letters, as they do not work until Monday morning. We dropped the letters strategically dropped on a weekend that had no rain forecasted to avoid damage to the letters. We dropped the letters approximately 5 m from a house driveway or front gate on the pedestrian walkway to ensure visibility. We did not drop any letters in front of any of the small businesses that exist in the residential suburbs, and also avoided construction sites. This increased the likelihood of the letters being returned by actual members of the area rather than short-term visitors. We did not drop any letters in sight of a post box or post office so as to make it more likely that the effort the finder would have to go to was roughly consistent across suburbs. There was only a maximum of one letter in each street to maximize the spread of the letters within the suburb, which reduced the likelihood of a participant coming across more than one letter and potentially alerting them to the nature of the experiment.

Ethics approval for the above project was granted in accordance with the requirements of the National Statement on Ethical Conduct in Human Research and the policies and procedures of The University of Western Australia (RA/4/1/7801).

### Data analysis

Suburbs were classified according their economic status. The Socio-economic Indexes for Areas (SEIFA) was used to determine the socioeconomic status of the different suburbs

in which the letters were distributed. Specifically, the Index of Relative Socioeconomic Disadvantage (IRSD) was used which ranks areas on a scale from most disadvantaged to least disadvantaged. The index takes into account 16 different variables from the 2011 census data, with each variable receiving a different weighting. Some of the more heavily weighted variables included the percentages of low-income houses, jobless parents, individuals living without internet and other variables including education level, occupation and average rent (*Australian Bureau of Statistics, 2009*; *Australian Bureau of Statistics, 2014*). These variables are combined to produce a decile ranking of deprivation for specific areas, on a scale of 1–10 (henceforth termed *socioeconomic index*). A score of 1 for an area shows that the residents in that area are in the most disadvantaged 10% in the nation. Some suburbs are composed of smaller statistical areas; for these suburbs the median rating of deprivation was taken (File S1). Suburbs characterized by large socioeconomic variation (difference between maximum and minimum IRSD for statistical areas >300) were not included in the experiment.

We first ran a Generalized Linear Mixed Model with binomial error structure and logit link function using the glmer function from the lme4 package (*Bates et al., 2015*) in R (*R Development Core Team, 2014*) version 3.1.0. The response variable–letter returned vs. not returned–was binary. Fixed effects were socioeconomic status, and whether or not a letter was stamped or unstamped. We also included number of postboxes in a suburb as a control variable. Suburb was classified as a random effect and included in the statistical model. Next, using a likelihood ratio test, we compared a saturated model containing all fixed effects with a null model containing none of the fixed effects but the same random effect as the saturated model (*Forstmeier & Schielzeth, 2011*). The interaction between stamped/unstamped and socioeconomic status was not significant and was therefore not retained in the final model. *P*-values for individual predictors were calculated based on Satterthwate's approximations using the lmerTest package (*Kuznetsova, Brockhoff & Bojesen Christensen, 2014*).

## RESULTS

A total of 92 stamped and 46 unstamped letters were returned (Files S1 and S2). A comparison of the full model to the null model showed that the set of predictors had a strong effect on whether a letter would be returned or not ($\chi^2 = 45.373$, $p < 0.001$). An analysis of the individual predictors in the model showed that unstamped letters had a significantly lower chance of being returned (estimate $= -0.320$, $SE = 0.054$, $p < 0.001$) (Fig. 1). Socioeconomic index also had a significant effect on whether or not a letter was returned (estimate $= 0.035$, $SE = 0.011$, $p = 0.00167$) (Fig. 2). A confounding effect of density of postboxes could be ruled out (estimate $= -0.002$, $SE = 0.012$, $p = 0.919$).

## DISCUSSION

A steady stream of recent research has undermined the original characterization of humans as *Homines economici* by uncovering substantial levels of prosocial behaviour (*Gintis, 2003*; *Henrich et al., 2004*; *Hill, Barton & Hurtado, 2009*; *Alvard, 2012*). The present field experiment using lost letters demonstrating people's willingness to engage in truly altruistic acts conforms with this paradigm. However, our experimental approach has revealed that

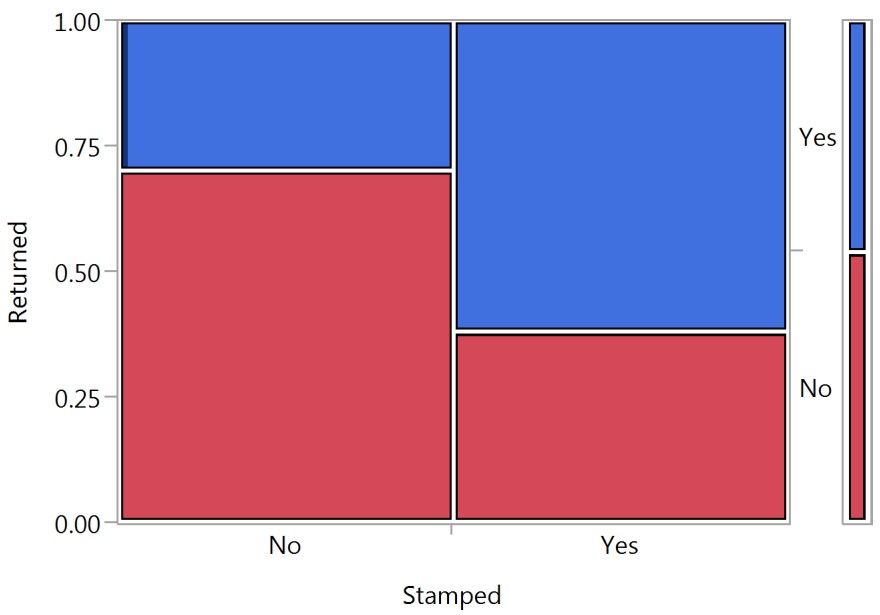

**Figure 1** Mosaic plot illustrating the percentage of returned letters as a function of whether they were stamped (Yes) or unstamped (No).

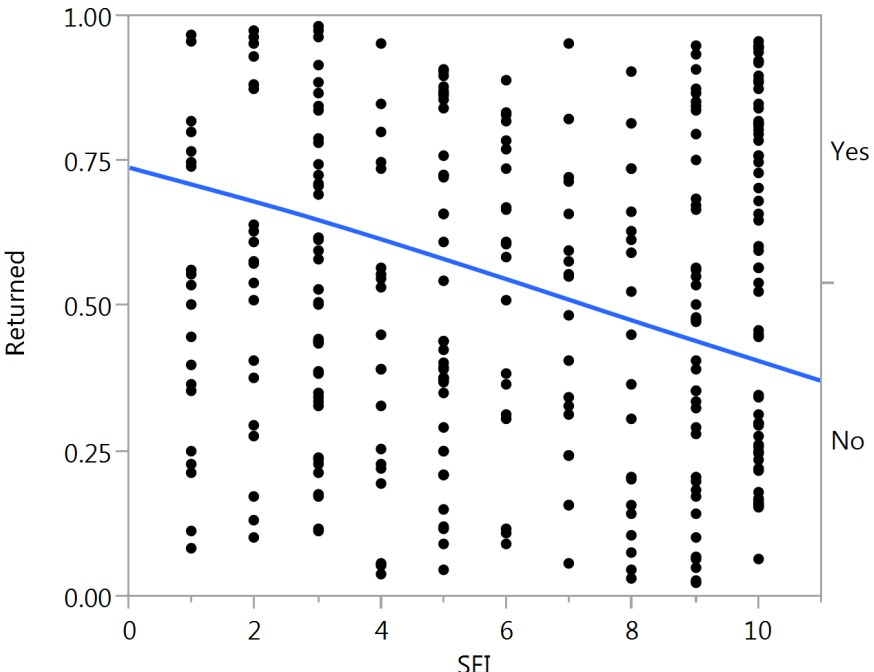

**Figure 2** Visualization of the effect of socioeconomic index (SEI) on whether a letter was returned or not (dichotomous variable). Letters dropped in high (10) SEI suburbs were more likely to be returned. The blue line represents a cut-point and not a trend line.

these altruistic tendencies vary strongly with both levels of neighborhood socioeconomic status and the costs involved in performing the altruistic act. Cost of the act has rarely been investigated within this experimental paradigm. When there was the likely added cost of going to a post office and purchasing a stamp, a letter was roughly half as likely to be returned. This is in line with results from economic games (*Isaac & Walker, 1988*; *Andreoni & Miller, 2002*), questionnaire-based studies on helping behaviour (*Stewart-Williams, 2007*) and two earlier implementations of the lost letter experiment (one of which was not couched in an evolutionary framework) (*Simon, 1971*; *Fessler, 2009*). The negative effect of low socioeconomic status on letter return rates is in agreement with most studies that utilized the lost letter technique to measure altruism.

The lower level of altruistic behaviour evident in poorer suburbs has been suggested to be a consequence of individual or neighborhood characteristics associated with socioeconomic deprivation (*Holland, Silva & Mace, 2012*). Individuals facing financial hardship, poor health and general life instability are likely to be preoccupied with achieving immediate needs, leaving less time and effort available to spend on benefiting a stranger (*Lynam et al., 2000*; *Holland, Silva & Mace, 2012*). In contrast, resource-rich individuals are not likely to be affected by such time and financial constraints. Individuals residing in poorer neighborhoods are also less likely to be embedded in a socially cohesive and supportive network and are exposed to higher levels of crime, conditions that discourage the development of trust required for civic efforts and prosociality (*Holland, Silva & Mace, 2012*). *Wilson, O'Brien & Sesma (2009)* found a good match between individual prosociality and the quality of the neighborhood (or more specifically the prosociality of the individuals' social environment) and reasoned that "this empirical result explains why it is possible for prosociality to succeed as a behavioural strategy in contemporary human life. Very simply, those who give to others also get from others" (p. 198).

One mechanism by which altruistic behaviour to unrelated individuals can be explained is reputation enhancement (*Nowak & Sigmund, 2005*). In our study, a number of the returned envelopes were annotated, detailing that the person had found and returned the letter on their own goodwill. In one instance, the mobile phone number of the finder was written on the envelope. In addition, one letter was hand delivered to the addressee's house. These actions suggest that the actors desired recognition of their good deed, supporting the theory of reputation enhancement.

Overall, our findings show that the willingness of individuals within a community to be altruistic decreases with increasing costs and social disadvantage. More broadly, this research shows that ecological variation within a given population can evoke divergent patterns of helping behaviour. In the context of business and industry, these results can aid charities and other crowd-funded organizations in directing their efforts to where they will likely receive the greatest return. Data such as the ones collected in this study provide a reflection of community attitudes and may therefore prove relevant to municipal government for policy development and intervention.

## ACKNOWLEDGEMENTS

We thank David Coall for helpful comments on this paper.

### Funding
The authors received no funding for this work.

### Competing Interests
The authors declare there are no competing interests.

### Author Contributions
- Cyril C. Grueter conceived and designed the experiments, performed the experiments, analyzed the data, wrote the paper, prepared figures and/or tables, reviewed drafts of the paper.
- Jesse A. Ingram, James W. Lewisson, Olivia R. Bradford, Melody Taba, Rebecca E. Coetzee and Michelle A. Sherwood conceived and designed the experiments, performed the experiments.

### Human Ethics
The following information was supplied relating to ethical approvals (i.e., approving body and any reference numbers):

Ethics approval for the above project was granted in accordance with the requirements of the National Statement on Ethical Conduct in Human Research and the policies and procedures of The University of Western Australia (RA/4/1/7801).

### Data Availability
The raw data has been supplied as Supplementary File.

### Supplemental Information
Supplemental information for this article can be found online at http://dx.doi.org/10.7717/peerj.2610#supplemental-information.

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
