# Peer review of "Human altruistic tendencies vary with both the costliness of selfless acts and socioeconomic status"

_PeerJ, doi:10.7717/peerj.2610_

## Round 0.1 · original submission · Major Revisions

Reviewer 1 argued the manuscript is inappropriate for PeerJ, given that the scope of the journal does not encompass papers that fall solely within the purview of the social sciences. I believe that you may be able to overcome this objection by elaborating on the relevant evolutionary issues in your Introduction. Please do so.

Reviewers 2 and 3 make important points regarding how the paper might be improved by better representing previous research and situating the present study within this literature. Please address these points.

Reviewer 3 makes several other important points as well. Please address them. Reviewer 3’s concern that the paper is not sufficiently novel is not valid given the stated scope of this journal (i.e., PeerJ assesses papers chiefly on the soundness of their Methods and Results, not the how interesting or novel the results seem to be.) However, this comment does again point out that you need to more clearly explain how these new results replicate or build upon previous ones.

Reviewer 4 makes several worthwhile suggestions for improving the writing in the annotated version that they uploaded. Please address them. They also make an important conceptual point, regarding deprivation ratings.

I have noted some Substantial Issues and some Minor Issues. Please address them.

Substantial Issues

Line 95 – you use the passive voice throughout the Methods, e.g., “letters…were dropped.” Please use the active voice and tell us who dropped the letters. Were they authors of the study? Could they have been aware of the hypotheses? My guess is that both of these things are true, and this seems likely to be a potential problem. In other words, if (sub-consciously) the letter-droppers wanted a higher return rate in an affluent neighborhood, they may have selected residences or neighborhoods that would lead to that outcome. So please rewrite the Methods using the active voice, disclosing more about the letter-droppers, and, if warranted, acknowledging (perhaps in the Discussion section) if there was a potential problem of the letter-droppers being aware of the hypotheses.

Line 138 – “A score of 1 for an area shows that the residents in that area are in the most disadvantaged 10% in the nation. There were numerous areas in each suburb, so the median rating of deprivation was taken from each suburb in our experiment (Appendix 1).”
Similar to Reviewer 4, I must ask “Why not use the actual ratings of the specific areas rather than median of their suburbs?” In fact, if your hypothesis is correct, the relationship shown in Figure 2 should improve when using areas. Please conduct this analysis or explain why it is not feasible.

Minor Issues

Line 116 - “This ensured the letters were returned by actual members of the area rather than short-term visitors.”
This practice would not “ensure” anything; it would merely increase the likelihood of them being returned by residents. Rewrite more precisely.

Line 117 - “Letters were not dropped in sight of a post box or post office so as to ensure the effort the finder would have to go to was consistent across suburbs.”
Again, this practice would not ensure what you are claiming. Be more precise and say something like “make it more likely that the effort finder would have to go to was ROUGHLY consistent across suburbs.”

Line 173 – “When there was the added cost of going to a post office and purchasing a stamp, a letter was roughly half as likely to be returned.”
Isn’t it possible that respondents had stamps on their person or at their residence and so would not necessarily need to go to a post-office? Please rewrite more precisely.

Line 209 – “Overall, our findings show that a community’s willingness to be altruistic decreases with increasing costs and social disadvantage.”
Be precise with your writing; you are assessing the behavior of INDIVIDUALS within a community, which is not same as the community.

Reviewer 1 ·

Basic reporting

Acceptable

Experimental design

UNACCEPTABLE -- The scope of the journal is defined with the following exclusion: "PeerJ does not publish in the Physical Sciences, the Mathematical Sciences, the Social Sciences, or the Humanities (except where articles in those areas have clear applicability to the core areas of Biological, Medical or Health sciences)." This paper unquestionably falls in the social sciences category. Granted, other authors working on the subject of prosociality have previously published papers in biological journals (such as Proc Royal Soc B), but, unlike this paper, those papers were explicitly framed in terms of evolutionary questions, importantly including the selection pressures thought to have been responsible for the evolution of prosocial behaviors unique to humans. The present paper does not make such connections. The authors would be better off sending this paper to a journal (such as PLoS ONE) that publishes social science research.

Validity of the findings

Acceptable

Additional comments

Nothing wrong with this paper, but it does not fall within the scope of this journal; the authors need to send it elsewhere.

·

Basic reporting

This appears to be a sound and well-written study. It replicates work that has already been done and largely concurs.

Minor point: the paragraph on confounds in the discussion (starting 197) should be removed. THe ideas discussed there are possible sources of noise, not confounds. I dont see the need to mention them.

Experimental design

The experimental design is sound.

Validity of the findings

The findings seems valid, although some of the framing I thought attacked a straw man, assuming some claim altruism being a universal should never vary (see below).

Additional comments

I dont think the statements made in the abstract and conclusions that altruism is a human universal and therefore finding variation in altruism is a new paradigm is a good way to frame this paper. Going back to Trivers and Hamilton we have always known that altruism is predicted to depend on cost and benefits. IT is true that some of the authors quoted have confused the issue and made unrealistic claims, but I dont think there is much need to get into that here. The paper is interesting and reporting an interesting poin t about wealth infuencing altruism, that concurs with Holland et al and Silva and Mace using a very similar method. There are one or two studies claiming the opposite. So it is good that this result is independently replicated. It does not necessary to claim that this is upsetting some wrong paradigm of universality that I dont think most people believed anyway.

·

Basic reporting

Introduction: The Introduction is a complete thought, but nonetheless very light on background literature. They do little to set up their hypotheses, framing them as either inductive (which would be fine if there weren't so much work in this area), or self-apparent (which would be surprising, again, given the amount of work in this area). They also make the claim that little has been done with within-society variation in levels of prosociality, a claim to which I'm sympathetic, but go on to list multiple studies that probe this exact question. They might want to be more precise in the ways these specific studies have and haven't handled this problem and what it is they are offering. There are also other studies they are overlooking as well.

Methods: The introduction of a new stimulus--lost letter without postage--is clever, but as the only clear original activity in the manuscript, I'm unclear on whether it is much of a contribution.

Results: The interpretation is pretty minimal here. Having to buy/contribute a stamp rather than just put a letter in a mailbox is obviously going to elicit less prosociality. The association between income and prosociality is not original either.

Discussion: The latter finding really needs further exploration. If the authors read the articles they cited upfront carefully, they are aware that the behavioral model is not "income-->prosociality" but that there are generally other social and contextual characteristics that encourage and inhibit prosociality in populations of different socioeconomic status.

Figure 2 is confusing. To the traditional dot-plot-trained eye, it looks like SEI predicts fewer returns, not more (because the blue line is interpreted not as a cut-point but as a trend line.

Experimental design

Fine, but limited in contribution. See above.

Validity of the findings

Reasonable enough, but see above regarding contribution.

·

Basic reporting

There are some issues related to the writing style and language which should be addressed. I point out some of these in the attached PDF, but I would recommend trying to improve the clarity of the writing in general.

Experimental design

Please clarify why was the median deprivation rating of each suburb used, instead of the deprivation rating of the area where the letters were dropped.

Validity of the findings

No Comments

---

## Round 0.2 · accepted · Accept

You've done a commendable job addressing my and the reviewers' concerns. This will make a fine contribution to the literature.